# Seasonal effects in the application of the MOMA remote calibration tool to outdoor PM<sub>2.5</sub> air sensors

4 Lena F. Weissert<sup>1</sup>, Geoff S. Henshaw<sup>1</sup>, Andrea L. Clements<sup>2</sup>, Rachelle M. Duvall<sup>2</sup>, and Carry Croghan<sup>2,3</sup>

- Aeroqual Ltd, 460 Rosebank Road, Avondale, Auckland, 1026, New Zealand; lena.weissert@aeroqual.com (L.F.W.);
- geoff.henshaw@aeroqual.com (G.S.H.)
- <sup>2</sup>U.S. Environmental Protection Agency, Office of Research and Development, Center for Environmental Measurement and
- Modelling, 109 T.W. Alexander Drive, P.O. Box 12055, Research Triangle Park, NC 27711; clements.andrea@epa.gov
- (A.L.C.); <u>duvall.rachelle@epa.gov</u> (R.M.D.)
- ³ retired; <u>carry.croghan@gmail.com</u> (C.C.)

3

5

- 13 Correspondence to: Lena F. Weissert (lena.weissert@aeroqual.com)
- Abstract. Air sensors are being used more frequently to measure hyper-local air quality. The PurpleAir sensor is among one
- of the most popular air sensors used worldwide to measure fine particulate matter (PM<sub>2.5</sub>). However, there is a need to
- understand PurpleAir data quality especially under different environmental conditions with varying particulate matter (PM)
- sources and size distributions. Several correction factors have been developed to make the PurpleAir sensor data more
- comparable to reference monitor data. The goal of this work was to determine the performance of a remote calibration tool
- called MOment MAtching (MOMA) for PM<sub>2.5</sub> sensors monitoring near temporally varying pollution sources of PM<sub>2.5</sub>. MOMA
- performs calibrations using reference site data within 0 15 km from the sensor. Data from 20 PurpleAir sensors deployed
- across a network in Phoenix, Arizona from July 2019 to April 2021. Results showed that the MOMA calibration tool made the
- PurpleAir PM<sub>2.5</sub> data more comparable to the co-located reference data (calibrated Mean Absolute Error (MAE): 2.8 3.7 µg
- m<sup>-3</sup>, Mean Bias Error (MBE): -1.8 0.1 µg m<sup>-3</sup>). The improvements were comparable to the Environmental Protection Agency
- (EPA) correction factor (MAE: 2.8 3.7 μg m<sup>-3</sup>, MBE: -0.9 0.4 μg m<sup>-3</sup>). However, MOMA provided a better estimate of
- daily average concentrations than the EPA correction factor when compared to the reference data under smoke conditions.
- Using the MOMA gain, representative of the sensor-proxy relationship, MOMA was able to distinguish between PM sources
- such as winter wood burning, wildfires, and dust events in the summer.

Key Words. Fine particulate matter, air sensors, PurpleAir, correction factors, MOMA, speciated PM

# 1 Introduction

particularly well-documented for PM<sub>2.5</sub>, with causal relationships established between both short- and long-term exposure and 34 respiratory, cardiovascular, and total mortality (U.S. EPA, 2019). Evidence also suggests that PM<sub>2.5</sub> poses a significant health 35 risk even at low concentrations with no 'safe' threshold (Elliott and Copes, 2011; Fann et al., 2012; U.S. EPA, 2019). Thus, 36 there is a need to measure air pollution at a high spatial and temporal resolution to understand localized air quality in areas 37 where people live, work, commute, and play (English et al., 2017). Data from lower-cost air sensors have great potential to 38 provide such insights helping communities to minimize exposure to unhealthy air pollution (Ardon-Dryer et al., 2019; Castell 39 et al., 2017; Snyder et al., 2013; Stavroulas et al., 2020). 40 The Plantower PMS 5003 (PMS, Plantower Technology, China) sensor, used in the PurpleAir sensor, is amongst the most 41 popular lower-cost sensor used to measure particulate matter (PM) concentrations. There are thousands of PurpleAir sensors 42 deployed globally providing publicly available data in real-time (https://map.purpleair.com/). The PMS sensor reports particle 43 counts between 0.3 and 10 μm across six size bins and provides mass concentration estimates for PM<sub>1</sub> (particles < 1μm in 44 diameter), PM<sub>2.5</sub> and PM<sub>10</sub> (particles < 10 um in diameter). Numerous studies have investigated the performance of PurpleAir 45 sensors in the field across the United States (U.S., Ardon-Dryer et al., 2019; Barkjohn et al., 2021; Feenstra et al., 2019; Jaffe 46 et al., 2023; Malings et al., 2020; Ouimette et al., 2022; Sayahi et al., 2019; Tryner et al., 2020b) and other parts of the world 47 (e.g., Greece [Kosmopoulos et al., 2022; Stavroulas et al., 2020], Australia [Robinson, 2020]) as well as in the laboratory 48 (Kuula et al., 2020; Tryner et al., 2020b). While these studies showed that the PurpleAir sensors are precise and linearly related 49 to PM<sub>2.5</sub> reference data up to approximately 250 µg/m<sup>3</sup>, they are not always accurate (Barkjohn et al., 2022). The measurement 50 range of PurpleAir sensors is notated to be from 0.3 – 10 µm, however, results suggest that the PurpleAir sensor is most 51 responsive to particles between 0.4 and 1 µm (Ouimette et al., 2024, 2022). Thus, concerns about the data quality in particular 52 under different environmental conditions with varying PM sources and size distributions remain (Barkjohn et al., 2021; Jaffe 53 et al., 2023; Ouimette et al., 2022; Williams, 2019). 54 As a result, recent work has focused on developing correction algorithms for PurpleAir sensors (Ardon-Dryer et al., 2019; 55 Barkjohn et al., 2021; Holder et al., 2020; Magi et al., 2020; Wallace et al., 2021, 2022). A U.S.-wide correction algorithm 56 developed by the U.S. Environmental Protection Agency's (EPA) showed significant improvements in accuracy across several regions and conditions (Barkjohn et al., 2021). The algorithm was further developed (Barkjohn et al., 2022) and is now 57 implemented in the AirNow Fire and Smoke map (https://fire.airnow.gov/). Despite the extensive dataset used in this study 58 59 the authors mention that further analysis may be required to test the performance of their correction long-term as well as for 60 extreme events and regions that were under-sampled but are characterized by different PM<sub>2.5</sub> sources (i.e., southern parts of the U.S., Northern Rockies, Ohio Valley). A recent study evaluating the performance of the correction for dust, wildfire, and 61 62 urban pollution suggested that the accuracy of the PMS using the Environmental Protection Agency (EPA) correction varies 63 for different pollution types (Jaffe et al., 2023). Additionally, the study found that the correction was not able to account for

Exposure to fine particulate matter ( $PM_{2.5}$ , particles 2.5 µm in diameter and smaller) is linked to adverse health effects. This is

64 missed particles associated with dust events and underestimated the reference PM<sub>2.5</sub> by a factor of 5-6. Wallace et al. (2022) developed a method to calculate PM<sub>2.5</sub> estimate from the particle count data and then established calibration factors using 65 reference sites within 500 m of the PurpleAir sensor for comparison. However, lower cost sensors are often deployed in areas 66 67 lacking monitoring data and consequently reference monitors are further away. 68 In the following work, we use a remote calibration tool, called MOment MAtching (MOMA), to calibrate PurpleAir sensors 69 deployed in Phoenix, Arizona. MOMA uses data from distant reference sites (within 0 – 15 km), referred to as 'proxies', to 70 determine if a sensor has drifted and to calculate the calibration gain and offset, referred to as MOMA gain and offset. MOMA 71 is based on the assumption that data from the reference site shows statistical similarity to those of the sensor site for a period 72 of time representative of the regional pollutant variation (Miskell et al., 2018, 2019; Weissert et al., 2020). MOMA has been 73 tested extensively in the Southern California region for ozone (O<sub>3</sub>), nitrogen dioxide (NO<sub>2</sub>), PM<sub>2.5</sub>, and PM<sub>10</sub> concentrations 74 from Aeroqual sensors (AQY, v1.0, Aeroqual Ltd, Auckland, New Zealand) (Miskell et al., 2019; Weissert et al., 2020, 2023). 75 Southern California is characterized by a Mediterranean climate with hot, mostly dry summers and mild winters. While 76 Phoenix, Arizona also experiences hot and dry summers, winter nighttime temperatures can go below freezing and smoke 77 related to wood burning is a common source of PM pollution, worsened by Phoenix's surrounding mountains which limit 78 vertical mixing of air pollutants (Grineski et al., 2007). In addition, Phoenix typically experiences 2 – 4 dust storms during its 79 monsoon season (June – September). The most intense dust storms, known as haboobs, move over Phoenix as a wall of dust 80 and may decrease the visibility to less than 1 km (Eagar et al., 2017). 81 The primary aim of this paper was to determine the performance of MOMA for PM<sub>2.5</sub> air sensors monitoring near temporally 82 varying pollution sources. Originally, MOMA was developed to correct sensor data associated with instrumental drift which 83 was defined as drift lasting for several days. However, previous research suggested that the MOMA calibration gain reflected changes in PM composition associated with different PM sources which can vary temporally from a few hours to days 84 85 (Weissert et al., 2023). Therefore, we are specifically interested in determining if a) MOMA can give insights into PM sources, 86 and b) if MOMA is able to detect sensor drift associated with different PM sources. The MOMA results are compared to the 87 EPA correction (Barkjohn et al., 2022).

#### 88 2 Methods

# **2.1 Datasets**

# 90 **2.1.1 PM<sub>2.5</sub> data**

- To better understand local air pollution associated with wood burning, Maricopa County and the U.S. EPA deployed a network
- of PurpleAir PA-II sensors across the Phoenix, Arizona Region between 2019 2021. From this network, we used data from
- four sites co-located with reference monitors (Mesa Brooks Reservoir, Durango Complex, South Phoenix, West Phoenix)

and 16 other sites with non-co-located PurpleAir sensors deployed across Phoenix (Figure 1). The network data are available from July 2019 – April 2021.

The PurpleAir PA-II contains two PMS sensors, referred to as 'Channel A' and 'Channel B'. The PMS sensor mass concentrations are either reported as cf\_1 or cf\_atm ('atmosphere'). For this analysis we use 'cf\_atm' data. PurpleAir raw data were reported at 2 min intervals and averaged to hourly values when data completeness for each hour was >75%. Hourly data from Channel A and Channel B were compared and data were removed when the hourly percentage difference between Channel A and B was > 70% and > 5  $\mu$ g/m³ (Barkjohn et al., 2021; Collier-Oxandale et al., 2022). The average PM<sub>2.5</sub> concentration from Channel A and Channel B was used for this analysis.

Figure 1. Map of PurpleAir sites and Reference sites in Phoenix. The speciation data were collected at PJLG, highlighted by the yellow star. The full names of the reference sites are shown in Table 1. The map was created using the Free and Open Source QGIS.

Hourly PM<sub>2.5</sub> reference data were accessed via AirNow (https://gispub.epa.gov/airnow/) (Figure 1). The reference network consisted of one Beta Attenuation Monitor (BAM) (BAM 1020, Met One Instruments, Inc., Grants Pass, Oregon, U.S.) at PJLG Supersite and seven dichotomous Taper Element Oscillating Microbalance (TEOM) monitors (TEOM 1405-DF, Thermo Scientific, Waltham, MA, U.S.). PurpleAir data were calibrated against the BAM and TEOM reference monitors. In addition, we used hourly PM<sub>2.5</sub> reference data from a reference grade optical instrument – T640x (Teledyne API, San Diego, U.S.), which was deployed at West Phoenix from November 2018 until April 2021 (the T640x data are not reflective of the April 2023 firmware update that implemented an alignment factor). The T640x data were used to assess whether the relationships observed between PurpleAir sensors and BAM/TEOM reference monitors across different PM sources also apply when using an optical reference instrument, they were not used to calibrate the sensor data at West Phoenix.

# 2.1.2 Supporting data

To determine if we can relate detected PM events to different PM sources, we used speciation data collected every third day at the PJLG Supersite (PJLG, Figure 1) and accessed via RAQSAPI package (McCrowey et al., 2022), which allows downloading monitoring data from the U.S. EPA Air Quality System (AQS) service. We focused on parameters representing crustal material (iron [Fe], magnesium [Mg], silicon [Si], potassium [K]), trace ions (potassium ion [K+], sodium ion [Na+], chloride [Cl-]), secondary ions (ammonium [NH4+], nitrate [NO3-], sulfate [SO42-]) elemental carbon (EC), and organic carbon (OC) following the PM source classification described in Weissert et al. (2023). Surface meteorological data from the Phoenix Sky Harbor International Airport were downloaded from the National Oceanic and Atmospheric Administration (NOAA) Integrated Surface Database (ISD) via the worldmet Package in R (Carslaw, 2022).

# 

# 2.2 Proxy selection

We selected proxies based on results from Weissert et al. (2023) which suggested that the nearest proxy (0-15 km) is generally a suitable choice for PM<sub>2.5</sub> (Table 1). The nearest proxy for the co-located sensor sites was between 5-6.5 km away from the sensor site and therefore representative of the sensor-proxy distance of most sites.

Table 1. Sensor site and the nearest proxy as well as the distance to the nearest proxy.

| Sensor site                 | Distance to nearest Proxy | Nearest Proxy                    | Co-located Proxy              |
|-----------------------------|---------------------------|----------------------------------|-------------------------------|
|                             | (km)                      |                                  |                               |
| West Phoenix                | 5.0                       | Phoenix JLG Supersite - PJLG     | West Phoenix - WP             |
| Durango Complex             | 4.9                       | South Phoenix - SP               | Durango Complex - DC          |
| Mesa – Brooks Reservoir     | 6.5                       | Tempe - TP                       | Mesa - Brooks Reservoir - MBR |
| South Phoenix               | 4.9                       | Durango Complex - DC             | South Phoenix - SP            |
| 33rd                        | 2.8                       | West Phoenix - WP                |                               |
| West 43rd                   | 3.3                       | Durango Complex - DC             |                               |
| Kyrene                      | 3.5                       | Tempe - TP                       |                               |
| Diablo                      | 3.6                       | Tempe - TP                       |                               |
| University                  | 4.3                       | Tempe - TP                       |                               |
| Highland                    | 4.8                       | Phoenix JLG Supersite - PJLG     |                               |
| Ocotillo                    | 4.8                       | Glendale Community College - GCC |                               |
| North 31st                  | 5.1                       | North Phoenix -NP                |                               |
| Stapley                     | 6.1                       | Mesa - Brooks Reservoir - MBR    |                               |
| Central Phoenix B           | 6.7                       | Phoenix JLG Supersite - PJLG     |                               |
| <b>Emergency Management</b> | 7.2                       | Tempe - TP                       |                               |
| Indian School               | 8.9                       | West Phoenix - WP                |                               |
| West Chandler               | 12.5                      | Mesa - Brooks Reservoir - MBR    |                               |
| Beardsley                   | 12.6                      | North Phoenix - NP               |                               |
| Indian Bend                 | 13.9                      | Tempe - TP                       |                               |
| Dysart                      | 15.8                      | Glendale Community College - GCC |                               |

# 135 **2.3 MOMA**

- PurpleAir data were corrected using our previously published MOMA calibration tool to detect sensor drift and apply a
- correction (Miskell et al., 2018, 2019; Weissert et al., 2020, Weissert et al., 2023). In brief, MOMA uses a two-sample
- Kolmogorov-Smirnov (K-S) test and the mean, E, -variance, var, (Mean-Variance, M-V) moment-matching MOMA gain,  $\hat{a}_1$
- and the intercept,  $\hat{a}_0$  to determine if sensor data, Y at location i, have drifted in relation to the proxy data, Z at location k, over
- the time interval  $t t_d$ : t.

$$\hat{a}_1 = \sqrt{var\{Z_{k,t-t_d:t}\}/var\{Y_{i,t-t_d:t}\}}$$
 (Eq. 1)

$$\hat{a}_0 = E\{Z_{k,t-t,q;t}\} - \hat{a}_1 E\{Y_{i,t-t,q;t}\}$$
 (Eq. 2)

- We used the following thresholds to determine sensor drift: K-S test p-value < 0.05, 0.75 >  $\hat{a}_1$  > 1.25; -5 µg m<sup>-3</sup> >  $\hat{a}_0$  > 5 µg
- m<sup>-3</sup>. We used a 3-day rolling average to calculate the statistics and a correction, as shown in Eq. 3, was triggered when any of
- the three statistics exceeded the threshold for a period of four consecutive days. The correction was applied retroactively to
- the start of this 4-day period.

$$\hat{C}_{PM_2} = \hat{a}_0 + \hat{a}_1 Y_{i,t}$$
 (Eq. 3)

- Subsequent corrections are only applied if the corrected PM<sub>2.5</sub>,  $\hat{C}_{PM_{2.5}}$ , have drifted in comparison to the proxy data. Given that
- we test the performance of MOMA for PM sources that may vary at shorter or longer timescales, we compare the MOMA
- corrected results from the 4-day threshold to those obtained using a 3-day threshold and 5-day threshold.
- The performance of the MOMA calibration tool was tested using PurpleAir data from South Phoenix, West Phoenix, Durango
- Complex and Mesa Brooks Reservoir calibrated against the nearest proxy reference to represent the impact of calibrating

against a distant proxy reference. The calibrated data were then compared against the co-located reference.

# 2.4 U.S.-wide PurpleAir correction

- To compare the results from MOMA we also applied the piecewise regression used for the AirNow Fire and Smoke Map to
- the PurpleAir data (Barkiohn et al., 2022):

if  $0 \le x \le 30$ :

$$y = 0.524x - 0.0862 \times RH + 5.75$$
 (Eq. 4)

if  $30 \le x < 50$ :

$$y = (0.786 \times (x/20 - 3/2) + 0.524 \times (1 - (x/20 - 3/2))) \times x - 0.0862 \times RH + 5.75$$
 (Eq. 5)

if  $50 \le x < 210$ :

$$y = 0.786x - 0.0862 \times RH + 5.75$$
 (Eq. 6)

if  $210 \le x < 260$ :

$$y = (0.69x \times (x/50 - 21/5) + 0.786 \times (1 - (x/50 - 21/5))) \times x - 0.0862 \times RH \times (1 - (x/50 - 21/5)) + 2.966 \times (1 - (x/50 - 21/5)) \times x - 0.0862 \times RH \times (1 - (x/50 - 21/5)) + 2.966 \times (1 - (x/50 - 21/5)) \times x - 0.0862 \times RH \times (1 - (x/50 - 21/5)) \times x - 0.0862 \times RH \times (1 - (x/50 - 21/5)) \times x - 0.0862 \times RH \times (1 - (x/50 - 21/5)) \times x - 0.0862 \times RH \times (1 - (x/50 - 21/5)) \times x - 0.0862 \times RH \times (1 - (x/50 - 21/5)) \times x - 0.0862 \times RH \times (1 - (x/50 - 21/5)) \times x - 0.0862 \times RH \times (1 - (x/50 - 21/5)) \times x - 0.0862 \times RH \times (1 - (x/50 - 21/5)) \times x - 0.0862 \times RH \times (1 - (x/50 - 21/5)) \times x - 0.0862 \times RH \times (1 - (x/50 - 21/5)) \times x - 0.0862 \times RH \times (1 - (x/50 - 21/5)) \times x - 0.0862 \times RH \times (1 - (x/50 - 21/5)) \times x - 0.0862 \times RH \times (1 - (x/50 - 21/5)) \times x - 0.0862 \times RH \times (1 - (x/50 - 21/5)) \times x - 0.0862 \times RH \times (1 - (x/50 - 21/5)) \times x - 0.0862 \times RH \times (1 - (x/50 - 21/5)) \times x - 0.0862 \times RH \times (1 - (x/50 - 21/5)) \times x - 0.0862 \times RH \times (1 - (x/50 - 21/5)) \times x - 0.0862 \times RH \times (1 - (x/50 - 21/5)) \times x - 0.0862 \times RH \times (1 - (x/50 - 21/5)) \times x - 0.0862 \times RH \times (1 - (x/50 - 21/5)) \times x - 0.0862 \times RH \times (1 - (x/50 - 21/5)) \times x - 0.0862 \times RH \times (1 - (x/50 - 21/5)) \times x - 0.0862 \times RH \times (1 - (x/50 - 21/5)) \times x - 0.0862 \times RH \times (1 - (x/50 - 21/5)) \times x - 0.0862 \times RH \times (1 - (x/50 - 21/5)) \times x - 0.0862 \times RH \times (1 - (x/50 - 21/5)) \times x - 0.0862 \times RH \times (1 - (x/50 - 21/5)) \times x - 0.0862 \times RH \times (1 - (x/50 - 21/5)) \times x - 0.0862 \times RH \times (1 - (x/50 - 21/5)) \times x - 0.0862 \times RH \times (1 - (x/50 - 21/5)) \times x - 0.0862 \times RH \times (1 - (x/50 - 21/5)) \times x - 0.0862 \times RH \times (1 - (x/50 - 21/5)) \times x - 0.0862 \times RH \times (1 - (x/50 - 21/5)) \times x - 0.0862 \times RH \times (1 - (x/50 - 21/5)) \times x - 0.0862 \times RH \times (1 - (x/50 - 21/5)) \times x - 0.0862 \times RH \times (1 - (x/50 - 21/5)) \times x - 0.0862 \times RH \times (1 - (x/50 - 21/5)) \times x - 0.0862 \times RH \times (1 - (x/50 - 21/5)) \times x - 0.0862 \times RH \times (1 - (x/50 - 21/5)) \times x - 0.0862 \times RH \times (1 - (x/50 - 21/5)) \times x - 0.0862 \times RH \times (1 - (x/50 - 21/5)) \times x - 0.0862 \times RH \times (1 - (x/50 - 21/5)) \times x - 0.0862 \times RH \times (1 - (x/50 - 21/5)) \times x - 0.0862 \times RH \times (1 - (x/50 - 21/5)) \times x - 0.0862 \times RH \times (1 - (x/50 - 21/5)) \times$$

$$(x/50 - 21/5) + 5.75 \times (1 - (x/50 - 21/5)) + 8.84 \times 10^{-4} \times x^2 \times (x/50 - 21/5)$$
 (Eq. 7)

if  $x \ge 260$ :

$$y = 2.966 + 0.69 \times x + 8.84 \times 10^{-4} \times x^2$$
 (Eq. 8)

174175

Where x is cf\_atm ( $\mu$ g m<sup>-3</sup>) and RH is relative humidity (%) measured by the PurpleAir Sensors.

# 2.5 PM<sub>2.5</sub> source identification

- Previous research suggested that the sensor-proxy relationship changes are associated with differences in PM sources and
- composition (Tryner et al., 2020; Jaffe et al., 2023; Ouimette et al., 2022; Weissert et al., 2023). Thus, we use the temporal
- variation of the MOMA gain,  $\hat{a}_1$  (Eq. 1), calculated over a 3-day rolling window ( $t_d$ ) to identify conditions when PurpleAir
- sensors tend to over- or underestimate PM<sub>2.5</sub> concentrations compared to the reference data. This MOMA gain is representative
- of the proxy relationship relative to the uncorrected PurpleAir data and independent of sensor drift alarms. The proxies outlined
- in Table 1 were also used to calculate the 3-day rolling MOMA gain.
- We assume that differences in the MOMA gain are, to some extent, related to the presence of different PM sources
- characterized by a different size distribution, chemical properties, or optical properties. In particular, we focused on
- distinguishing between smoke, dust, and general urban traffic PM sources. These sources are common in Phoenix and are
- associated with different particle sizes and composition (Jaffe et al., 2023). We averaged the MOMA gain for each day to
- identify days dominated by smoke, dust, and typical urban PM sources. We then tested the performance of MOMA and the
- EPA correction under these different conditions.

# 3 Results and Discussion

199

202

207

212

#### 3.1 Seasonal effects and PM source identification

5 6 10

Figure 2 shows the monthly variations of the MOMA gain used to correct the PurpleAir sensor data across the whole dataset and the number of drift alarms raised at each site. Figure 2a indicates that the MOMA calibration gain represents monthly variations that are more consistent with changes in PM sources rather than with instrumental drift. Unlike most other sites, the drift detection test triggered frequent alarms at 33rd across the whole study period despite being close to the proxy site (2.8 km), Mean uncorrected PM<sub>2.5</sub> concentrations across the whole study period at 33<sup>rd</sup> were 4 ug m<sup>-3</sup>, less than half of those recorded at West Phoenix (10 µg m<sup>-3</sup>). Low concentrations at 33<sup>rd</sup> were surprising given the site is located next to a busy motorway, indicating a potential sensor or siting issue for a PM<sub>2.5</sub> measurement at this site (Figure S1). Alternatively, frequent alarms could signify an unsuitable proxy match due to different particle sources. The 33<sup>rd</sup> site is influenced predominantly by vehicle traffic and the West Phoenix site is influenced by the residential community surrounding the site with particle sources like lower-speed road dust, cooking emissions, and residential smoke (e.g., domestic heating, backyard fire pits). Similarly, Dysart resulted in frequent calibrations which is likely a result of an unsuitable proxy given the site is located north of Phoenix the furthest away from any reference site (> 15 km) and selected as a background site.

Figure 2. a) Boxplot showing the monthly variations of the MOMA gain from the drift detection framework used to correct the PurpleAir sensor data across the whole network coloured by season. The numbers at the top indicate the number of outliers observed for different months, b) Bar chart showing the number of drift alarms and calibrations triggered at each site.

MOMA only triggers a calibration when a sensor shows a statistical difference for a continuous period of four days. While this is suitable for long lasting pollution events or instrumental sensor drift, it risks missing short-term events such as dust storms or PM sources that show diurnal variations such as wood smoke related to domestic heating. Additionally, the MOMA gain produced by the calibration tool and shown in Figure 2a is calculated relative to the previously applied MOMA gain and offset, meaning it reflects changes in the sensor-proxy relationship based on prior corrections rather than an absolute comparison to the proxy reference. Thus, the 3-day rolling MOMA gain from July 2019 – April 2021, shown in Figure 3a, provides more detailed insights into the temporal variability of PM sources independent of the drift detection framework and representative of the relationship between the uncorrected PurpleAir data and the proxy reference data. Figure 3a clearly shows the observed MOMA gain < 1 for PurpleAir sensors during the winter and occasionally during the summer (e.g., August/September 2020), suggesting PurpleAir sensors overestimate PM<sub>2.5</sub> compared to the proxy reference under these conditions. Two factors may contribute to the PurpleAir sensors overestimating in winter, particularly at night and in the early morning hours (8 pm - 7 am, Figure S2). First, RH is higher at night as well as during winter (Figure 3b), which can lead to hygroscopic growth of particles impacting the light scattering (Badura et al., 2018; Morawska et al., 2018). Second, other research (e.g., Jaffe et al., 2023; Barkjohn et al., 2021a; Holder et al., 2020; Tryner et al., 2020a) found that PurpleAir sensors overestimated in the presence of smoke particles and typical secondary urban aerosols. The difference between the PurpleAir sensor and the reference data varies diurnally over winter with the largest differences observed at night and in the early morning (Figure S2). Wood burning is more popular during winter evening recreation. It is therefore possible that wood smoke, a dominant PM<sub>2.5</sub> source in winter in Phoenix (Ramadan et al., 2000), accumulates at night and that colder nighttime temperatures and overnight winter inversions can lower the mixing height thereby trapping emissions and elevating PM<sub>2.5</sub> concentrations contributing to a lower MOMA gain in winter. In contrast, there are periods (most frequently in July and August) when the PurpleAir sensors underestimate compared to the proxy reference (MOMA gain > 2.5) (Figure 3). These periods were generally short and easily seen as PM<sub>2.5</sub> spikes in the reference data as well as a reduction in visibility measured at the airport. According to White et al. (2023) a major dust storm was recorded in the NOAA Storm Event Database on August 16, 2020 in Maricopa agreeing with the spike from the reference data (White et al., 2023). However, this was not observed in the sensor concentrations (e.g., August 16-17, 2020) consistent with previous findings that PurpleAir sensors underestimate PM<sub>2.5</sub> concentrations during dust events due to the narrower particle size range detected (Ouimette et al., 2022; Stavroulas et al., 2020). Shortly after the dust event the 3-day rolling MOMA gain was 

Figure 3. a) Daily averaged MOMA gain across the entire sensor network from July 2019 - April 2021. The daily values represent the mean of a 3-day rolling average applied to the hourly dataset, the shaded area is the standard deviation, b) relative humidity measured at the Phoenix Sky Harbor International Airport.

Given our findings, we separated our dataset into MOMA source categories representing smoke (MOMA gain  $\leq$  0.6) and dust conditions (MOMA gain > 2.5) with the remaining data considered to be typical urban PM<sub>2.5</sub> (traffic, industrial). This resulted in 25 dust days (all in summer) and 16 smoke days (winter: 12, spring: 2, summer: 2) and 555 days dominated by typical urban conditions between July 2019 and April 2021. Figure 4 shows the mean reference PM<sub>2.5</sub>/PM<sub>10</sub> ratio for each MOMA source category. As expected, smoke days were associated with a higher reference PM<sub>2.5</sub>/PM<sub>10</sub> ratio (mean = 0.6) and with higher concentrations of elemental carbon (EC) and organic carbon (OC) compared to the other two categories which is also observed by Tong et al. (2012). Dust days, on the other hand, were dominated by larger particles (PM<sub>10</sub> = 13.7 – 126.4  $\mu$ g m<sup>-3</sup>, PM<sub>2.5</sub> = 3.7 – 18.4  $\mu$ g m<sup>-3</sup>) and higher concentrations of crustal elements compared to EC and OC concentrations. As a result, the PM<sub>2.5</sub>/PM<sub>10</sub> ratio (mean = 0.2) was considerably lower compared to the smoke days. It is important to note that speciation data were only collected every third day and only nine out of the 25 sampling days contained a full complement of data. Other sampling days were missing some speciation data. The urban category showed a more uniform distribution of species concentrations. Some urban days may coincide with short-lived dust and smoke events, and applying the daily averaged MOMA gain across the entire network — along with the choice and representativeness of proxies and thresholds — introduces

classification uncertainties. The light grey shaded area in Figure 3a represents the standard deviation of the daily MOMA gain, suggesting that the gain varies across sensors and hours within each day. This variability likely reflects differences in environmental conditions and the sensor-reference relationship across the network. Nevertheless, the  $PM_{2.5}/PM_{10}$  ratios are in agreement with ratios (0.15 – 0.26) used by the U.S. EPA for fugitive dust particles (Midwest Research Institute (MRI), 2005) and other studies (Tong et al., 2012: 

Figure 4. a) Daily averaged PM<sub>2.5</sub>/PM<sub>10</sub> reference ratio measured across the AirNow reference network grouped by the three MOMA source categories, and b) species concentrations, normalized to the mean category (Crustal Elements, EC, OC, Secondary Ion, Trace Ion) values, measured at PJLG for the three different MOMA source categories.

# 3.2 Performance of MOMA under different conditions

Figure 5 shows the co-located reference data against the PurpleAir data without any correction (a), with the EPA correction (b), and the MOMA calibration against the nearest proxy reference (c) for data from the four co-location sites (Durango Complex, West Phoenix, South Phoenix, Mesa – Brooks Reservoir). As previously observed (Barkjohn et al., 2021; Holder et al., 2020; Jaffe et al., 2023), uncorrected PurpleAir sensor data correlated well with the co-located reference data as indicated by the high  $R^2$  (>0.7) for winter smoke and typical urban conditions but overestimated PM<sub>2.5</sub> at higher concentrations when smoke particles are present and RH is high (mean bias error [MBE] = -7  $\mu$ g m<sup>-3</sup>). As expected, the relationship between the PurpleAir sensors and the co-located reference is poorer during dust conditions ( $R^2 = 0.53$ ) and the PurpleAir sensors underestimated PM<sub>2.5</sub> (MBE = 3.1  $\mu$ g m<sup>-3</sup>). Both the EPA correction and the MOMA calibration resulted in improved accuracy

for all source categories as indicated by a reduction in the mean absolute error (MAE) and the MBE. Overall, MOMA 280 performed comparably to the EPA correction in terms of MAE (2.8 – 3.7 µg m<sup>-3</sup> for both) and root mean square error (RMSE) (4.8 – 6.2 µg m<sup>-3</sup> vs. 4.5 – 7.6 µg m<sup>-3</sup>, respectively) (Figure 5). MOMA performed slightly better correcting for dust conditions, 281 282 but neither the EPA correction nor the drift calibration framework were able to fully correct for the missed dust particles at 283 high concentrations and the RMSE remained high. This was also observed in Figure 6 which shows uncorrected, MOMA 284 calibrated and EPA corrected data from August 2020 when a dust event (Figure 6a) moved over Phoenix followed by wildfire 285 smoke (Figure 6b). Both the MOMA calibrated data and the EPA corrected data missed the dust spike but performed well 286 during the wildfire event. In contrast, the T640x detected the dust event indicating it is sensitive to the larger dust particles. 287 This highlights the limitations of the design of the lower cost PM sensor in the PurpleAir sensor which is insensitive to crustal 288 dust (Jaffe et al., 2023; He et al., 2020; Ouimette et al., 2022; Kuula et al., 2020). This may be due to poor transport of larger 289 particles into the sensor during high wind, into its optical cavity, and/or poor light scattering design (Kuula et al., 2020; 290 Ouimette et al., 2024). In fact, mean wind speeds were higher during dust days (mean = 3.5 m s<sup>-1</sup>) compared to the other two 291 MOMA source categories (< 3 m s<sup>-1</sup>), supporting findings by Ouimette et al. (2024), who reported a decline in sampling 292 efficiency for larger particles at wind speeds greater than 3 m s<sup>-1</sup>. In contrast, during the wildfire episode (Figure 6b) both the 293 T640x and the PurpleAir sensors showed a positive bias versus the TEOM. This is well known for PurpleAir sensors (Barkjohn 294 et al., 2021; Jaffe et al., 2023) and has also been observed for the T640x (Barkjohn et al., 2022; Long et al., 2023). 295 Table 2 shows the number of days concentrations detected by the reference instrument were above the National Ambient Air Ouality Standard (NAAOS) primary 24-hour PM<sub>2.5</sub> standard (35 µg m<sup>-3</sup>) as measured by the uncorrected sensor data, the 296 297 MOMA corrected data against the nearest proxy, and the EPA corrected data. The reference instruments did not report 298 concentrations above this threshold under smoke or dust conditions but did so on seven days under typical urban conditions. 299 Both the MOMA and EPA corrections improved accuracy near this concentration threshold but still slightly overestimated 300 concentrations under urban conditions, with MOMA performing slightly better than the EPA correction.

Figure 5. Hourly sensor measurements compared to reference data at the four co-location sites for a) uncorrected ( ${}^{\circ}PM_{2.5}$ \_atm')  $PM_{2.5}$  sensor data, b) corrected sensor data using the EPA correction, and c) MOMA calibrated sensor data using the nearest proxy reference across the MOMA source categories (Dust: 1992 hours, Smoke: 1251 hours, Urban: 44601 hours). Points are colored by RH (%).

Figure 6.  $PM_{2.5}$  concentrations measured at West Phoenix by the co-located TEOM and T640x reference and the PurpleAir Sensor (PM2.5 ATM – Uncorrected), with the EPA correction and MOMA applied during the a) dust event, and b) the wildfire smoke conditions.

Table 2. Number of times the daily average concentration went above the EPA primary 24-hour PM2.5 standard (> 35  $\mu g$  m<sup>-3</sup>) detected by the co-located PurpleAir sensors when using uncorrected, MOMA calibrated against the nearest proxy, and the EPA corrected data across different MOMA source categories.

|                      | PurpleAir Sensors |      |                       |           |  |
|----------------------|-------------------|------|-----------------------|-----------|--|
| MOMA source category | Uncorrected       | MOMA | <b>EPA</b> correction | Reference |  |
| Dust                 | 0                 | 0    | 0                     | 0         |  |
| Smoke                | 5                 | 0    | 0                     | 0         |  |
| Urban                | 42                | 12   | 14                    | 7         |  |

Table 3 shows a comparison of the MAE for MOMA with a 4-day (used throughout this paper), 3-day, and 5-day threshold to raise a drift alarm for each MOMA source category at the co-location sites. The length of the window used to trigger a calibration is a trade-off between over-correction and removal of local effects (shorter window), particularly when calibrating against a distant proxy, and minimising the response time if a sensor is drifting (longer window). This is reflected in Table 3, which shows best improvements with a 4-day threshold and a decrease in the improvement of the MAE using a shorter or longer window to trigger a calibration, although there is some variation depending on the MOMA source category. This was observed across all sites however the improvement was slightly better at West Phoenix and Mesa – Brooks Reservoir. The sensor – proxy pair South Phoenix – Durango Complex does not appear to be a suitable match. Durango Complex is located next to the City of Phoenix Transfer Station, a garbage collection site, and the pollutant sources as well as temporal variations may therefore be different from those observed at South Phoenix, which is representative of a residential area. However, overall differences are small and the length of the window to trigger an alarm may also depend on the MOMA source category and its spatial variability.

Table 3. Average percent improvement in the MAE before (PM<sub>2.5</sub> ATM) and after the MOMA calibration for different drift detection thresholds measured for PurpleAir sensors at the co-location sites West Phoenix, South Phoenix, Durango Complex and Mesa – Brooks Reservoir. Data were calibrated against the nearest proxy reference site and results compared to the co-located reference. These thresholds determine when a drift alarm is raised (i.e., the number of consecutive days a warning needs to be exceeded for before a correction is triggered).

| % improvement in the MAE   |        |        |        |  |  |  |
|----------------------------|--------|--------|--------|--|--|--|
| MOMA<br>source<br>category | 3 days | 4 days | 5 days |  |  |  |
| Dust                       | -5     | 24     | 22     |  |  |  |
| Smoke                      | 59     | 54     | 58     |  |  |  |
| Urban                      | 24     | 26     | 26     |  |  |  |

# 4 Conclusions

This paper provided insights into the performance of the MOMA remote calibration tool and EPA correction for PurpleAir sensors and highlighted challenges related to calibrating PM sensors for seasonally and diurnally varying PM sources. MOMA improved the accuracy of the PurpleAir data resulting in a similar hourly MAE, MBE and RMSE as the EPA correction. The MOMA gain also provided a simple method of distinguishing between different MOMA source categories such as wood burning in winter and wildfires and dust events in summer. This may help identify PM sources based on sensor-reference

- relationships. However, further analysis is required to test the application of this classification across other climate regions in
- the U.S. or worldwide and to refine thresholds used to classify dust and smoke events and to distinguish these efficiently from
- typical urban conditions.
- We have highlighted the challenges in the calibration of PurpleAir sensors for Phoenix conditions in which the magnitude and
- sources of PM<sub>2.5</sub> vary. The EPA correction is independent from reference sites and works well for smoke dominated conditions
- but poorly for dust events. MOMA, being a dynamic method, is better at accommodating PM source variations but rapid,
- short-term changes such as dust events and diurnally varying woodsmoke patterns can also be challenging. In addition, MOMA
- is dependent on reliable proxy reference sites within 15 km to the sensor site representative of average pollution sources present
- at the sensor site and the drift detection framework as applied here relies on several days of data to determine if a sensor has
- drifted.

- Funding: The measurement campaign was funded internally by the U.S. Environmental Protection Agency's Office of
- Research and Development. In-kind field support was offered by Maricopa County Air Quality Department. In-kind analysis
- support was offered by Aeroqual Ltd.

- Author contributions: G.S.H., R.M.D. and A.L.C. formulated the overarching research goals and aims; A.L.C. planned and
- implemented the field study; L.F.W. and G.S.H. developed the methodology for this analysis; C.C. organized and quality
- assured the dataset used for this analysis; L.F.W. developed the software and performed the data analysis; L.F.W. prepared the
- manuscripts with contributions from all co-authors; R.M.D., A.L.C. and G.S.H. supervised the project.

- **Disclaimer**: This document has been reviewed in accordance with the U.S. Environmental Protection Agency policy and
- approved for publication. Mention of trade names or commercial products does not constitute endorsement or recommendation
- for use. The views expressed in this journal article are those of the authors and do not necessarily reflect the views or policies
- or the U.S. Environmental Protection Agency.

- Competing interests: The authors declare the following financial interests/personal relationships which may be considered as
- potential competing interests: L.F.W. and G.S.H. are employees of Aeroqual Ltd, a sensor manufacturer. G.S.H. is a founder
- and shareholder in Aeroqual Ltd.

- Code and Data availability: The PurpleAir data from the Phoenix network are archived on Zenodo at
- https://zenodo.org/records/15174476 (Clements et al., 2025). The dataset used to explore seasonal effects in the application of
- the MOMA remote calibration tool is accessible via <a href="https://doi.org/10.23719/1532147">https://doi.org/10.23719/1532147</a> (Clements, n.d.). The code is not
- publicly accessible due to intellectual property.

- Acknowledgements: The authors acknowledge Sue Kimbrough (EPA retired), Ben Davis, Ron Pope, and Ira Domsky
- (Maricopa County Air Quality) for field study planning and implementation, Tim MacArthur, Alex Korff, Jacob Cansler, and
- Parik Deshmukh (Jacobs) for designing and building the P-TAQS pods, field data collection support by Robert Dyer (Maricopa
- County), data tracking and transfer by Nikki Petterson, Ceresa Steward, Hirna Patel (Maricopa County) and Brittany Thomas
- (Jacobs), and data quality assurance and preliminary data analysis by Ian VonWald (former ORISE postdoc to EPA).

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
