# Peer review of "Seasonal effects in the application of the MOMA remote calibration tool to outdoor PM2.5 air sensors"

_EGUsphere, 2024_

## Referee Comment (RC2)

"Seasonal effects in the application of the MOMA remote calibration tool to outdoor PM$_{2.5}$ air sensors"

**General Comments**

The manuscript presents research work using a remote calibration method (Moment Matching, MOMA) for a network of PurpleAir PM$_{2.5}$ measurements made in Phoenix, Arizona USA between July 2019 and April 2021. While the authors showed that the MOMA approach compares well with compared the performance of the MOMA approach with US EPA correction methodology in improving the accuracy of the PM$_{2.5}$ data across the PurpleAir network, the presented evidence that MOMA performed better in estimating daily exceedance of smoke levels compared to the EPA correction when both methodologies were compared to exceedances derived from reference data.

Finally, the authors further showed that they could identify different PM$_{2.5}$ categories corresponding to dust, smoke and urban aerosol by using information from the MOMA gains estimated from 3-day rolling average, the PM categories were assessed using reference PM speciated measurements.

**Specific comments**

The authors have used specific MOMA gain threshold for the PM source category identification (MOMA gain < 0.6 ≡ smoke, MOMA gain > 2.5 ≡ dust and 0.6 < MOMA gain < 2.5 ≡ urban) based on the network averaged MOMA gains derived for the PurpleAir. It would be good to add some uncertainty to criteria. I would suggest the authors not only show the average daily MOMA gains across the network (Figure 3) but also include the standard deviation of the daily MOMA gains across the network as a measure of confidence in the derived daily MOMA gains. This is particularly important as the authors have indicated in Figure 2 that there might be sites that trigger frequent MOMA calibrations like Dysart and how does this affect the statistics (mean MOMA gain) presented in Figure 3.

Technical corrections

P.2, line 41: PM$_{2.5}$ size range is not defined as was done for PM$_1$ and PM$_{10}$.

P. 4, line 96: add that 70% is actually '70% hourly percentage difference'.

P. 4, line 11: authors need to specify that the number of JLG Supersites is two.

P. 7, line 178: Author needs to include the period over which the "Mean uncorrected PM2.5 concentration..." was estimated over. Was it for the entire duration of the deployment?

P.9 line 201, Looking at the interpretation of Figure 3 and the MOMA gain thresholds, I would say the short periods with MOMA gains > 2.5 are mainly June and August.

P.9 line 215, "change the PM$_{2.5}$/PM$_{10}$ for each PM$_{2.5}$ source category" to "the reference PM$_{2.5}$/PM$_{10}$ for each PM$_{2.5}$ source category identified by MOMA"

P.10 line 224, change "source categories, and b)…" to "MOMA source categories, and b)…"

P.10 line 224,   change "… b) normalized species concentrations measured for….."  to "… b) normalized species concentrations measured at two Supersites for …"

P.12 line 260,   change "Nr. " to "Number"

P.14 line 303-305,   A.D. is not listed in the author list, remove? Also, C.C is missing from this list.

P.14 line 312-314,   Keep the abbreviation consistent, G.S.H should read G.H.  (see lines 303-305). D.E.W. is not an author in this manuscript, delete or include if missing.

---

## Author Response (AR1)

We thank the reviewer for the constructive and helpful comments. The reviewer's comments were addressed point-by-point. Our response is highlighted in red below and in yellow in the manuscript.

Response to Reviewer 1

Line 35-38: Based on your statement, I'm curious about how the correction process is applied. If the 3-day rolling average triggers a correction after exceeding the threshold for 4 consecutive days, would the correction only apply to data collected from Day 4 onward, or would it retroactively adjust the data for Days 1, 2, and 3 as well? Yes, the correction was applied retroactively to adjust the data for Days 1,2 and 3 as well. This clarification has been added (L147).

Line 111: Could you clarify whether the speciation data is collected every third day? If so, does this limit the number of days labeled as "Dust" that actually have corresponding speciation data? Given that dust events are short-lived, does the speciation data reveal any significant differences in crustal composition between days labeled as "Dust" and "Urban"? In Figure 4, the comparison of crustal composition shows little difference between the means for Dust and Urban. Could this be influenced by the disparity in the number of days analyzed (20 Dust days vs. ~600 Urban days)? Thank you for pointing this out, speciation data were collected every third day (L119) and therefore not all 'dust' days had speciation data. We have added a discussion about limitations and uncertainties related to the speciation data as well as the MOMA source categories (L252 – 259).

Lines 103–106 state: "The reference network consisted of 1 Beta Attenuation Monitor (BAM) at the Phoenix JLG Supersite and 8 dichotomous Taper Element Oscillating Microbalance (TEOM) monitors." Does this mean that the Phoenix JLG Supersite has both BAM and a TEOM monitor? Apologies, this was an error, there are only 7 TEOMs, PJLG only has a BAM.

Line 107-108: The authors mention: "For comparison, we also used hourly PM2.5 reference data from a reference-grade optical instrument – T640x (Teledyne API, San Diego, U.S.), which was deployed at West Phoenix from November 2018 until April 2021." Could you clarify what specific comparison the authors are referring to in this context?" We have added some further detail on L114. The T640x data were only used for the comparison in Figure 6.

Line 119 – 121: If multiple proxy sites are equidistant from a specific PurpleAir sensor, which one did the authors choose for comparison? Were there instances where these two proxy sites showed different concentrations? Each site used the nearest proxy reference, and we did not complete proxy comparisons as part of this manuscript as a previous publication (Weissert et al., 2023) addressed the selection process of proxies. If there were sites with equidistant proxies one may want to review other site information to choose the most suitable proxy.

Line 128: Please specify what does MV stands for? MV stands for Mean-Variance, we clarified that on L138.

Line 128: Please check if "E – variance, var (MV)" should be "E, variance, var"? Thank you, this is now correct (L138).

Line 139-141: The author mentions: "The impact of this threshold was tested using Purple Air data from South Phoenix, West Phoenix, and Durango and calibrating these against the nearest proxy reference, to represent the impact of calibrating against a distant proxy reference." Are the authors emphasizing that these sites had collocated sensors, but the calibration was still

performed using the nearest proxy reference? Yes, correct. In fact, we decided to update the whole manuscript to calibrate all sensors, even those with a co-located reference, against the nearest Proxy, we have also added the 4th co-location site (Mesa – Brooks Reservoir) to this comparison. The performance of the calibration was then completed against the co-located reference if available (Table 2, L155). This allowed for a fairer comparison against the EPA correction and is more representative of the performance of MOMA at sites with no co-located reference data available. As a result, the MOMA source categories and results associated with these changed slightly in the revised version.

Line 161: About RH, is the PurpleAir sensor reported relative humidity or meteorological data? RH was used from the Purple Air sensors (L176).

Line 168: Please correct "between smoke and dust and general urban traffic PM sources" to "between smoke, dust, and general urban traffic PM sources" Done (L187).

Figure 2: Please specify if it is 3-day rolling MOMA gain daily average. Figure 2 has changed for improved clarity and the MOMA gain shown in Figure 2 is described on L214.

Line 194-195: The authors mention that "The difference between the PurpleAir sensor and the reference data varies diurnally over winter with the largest differences observed at nighttime." Since diurnal data is not presented in the manuscript, please clarify where this information is derived from. A Figure supporting this has been added to the Supporting Information (Figure S2).

Figure 4: Were the species concentrations normalized by the maximum concentration? Could you please specify how the normalization was performed? These were normalised against the mean Category (Crustal Elements, EC, OC, Secondary Ion, Trace Ion) values (L267).

Figure 4: Did the remaining ~600 days (between July 2019 and April 2021, excluding 20+18 for Smoke and Dust) contribute to Urban days? Yes, these are considered 'urban days'. This has been added to the revised manuscript (L247).

Figure 4: Please provide the range of PM10 concentrations from the regulatory monitor for the marked "Dust" days. A low PM2.5/PM10 ratio does not necessarily indicate high absolute PM10 (observed during Dust events). For example, PM10 concentrations below 50 µg/m³ can also result in low PM2.5/PM10 ratios. That is correct, however, the range of PM10 concentrations for dust days was indeed larger. We have added some further information (L251).

General comment: Should the authors accept as a limitation that their method does not allow real-time correction, unlike the EPA's method that uses the PurpleAir parameter for real-time data correction? We have added this as a limitation (L349). In the context of this paper and using the drift detection framework MOMA is relying on some historical data, however, a previous publication (Weissert et al., 2023) also shows that it can be applied in near real-time.

**General Comments**

The manuscript presents research work using a remote calibration method (Moment Matching, MOMA) for a network of PurpleAir $PM_{2.5}$ measurements made in Phoenix, Arizona USA between July 2019 and April 2021. While the authors showed that the MOMA approach compares well with compared the performance of the MOMA approach with US EPA correction methodology in improving the accuracy of the $PM_{2.5}$ data across the PurpleAir network, the presented evidence that MOMA performed better in estimating daily exceedance of smoke levels compared to the EPA correction when both methodologies were compared to exceedances derived from reference data.

Finally, the authors further showed that they could identify different $PM_{2.5}$ categories corresponding to dust, smoke and urban aerosol by using information from the MOMA gains estimated from 3-day rolling average, the PM categories were assessed using reference PM speciated measurements.

**Specific comments**

The authors have used specific MOMA gain threshold for the PM source category identification (MOMA gain < 0.6 ≡ smoke, MOMA gain > 2.5 ≡ dust and 0.6 < MOMA gain < 2.5 ≡ urban) based on the network averaged MOMA gains derived for the PurpleAir. It would be good to add some uncertainty to criteria. I would suggest the authors not only show the average daily MOMA gains across the network (Figure 3) but also include the standard deviation of the daily MOMA gains across the network as a measure of confidence in the derived daily MOMA gains. This is particularly important as the authors have indicated in Figure 2 that there might be sites that trigger frequent MOMA calibrations like Dysart and how does this affect the statistics (mean MOMA gain) presented in Figure 3. Thank you for this suggestion, we have added the daily standard deviation to Figure 3 and further discuss uncertainties related to the identified MOMA source categories (L253-L260).

Technical corrections

P.2, line 41: $PM_{2.5}$ size range is not defined as was done for $PM_1$ and $PM_{10}$. The size range for PM2.5 is defined earlier on L32.

1. 4, line 96: add that 70% is actually '70% hourly percentage difference'. Done (L99).

2. 4, line 11: authors need to specify that the number of JLG Supersites is two. This has been clarified in Figure 1 as well as L108 – L116.

3. 7, line 178: Author needs to include the period over which the "Mean uncorrected PM2.5 concentration..." was estimated over. Was it for the entire duration of the deployment? Yes, this was across the whole study period. We have clarified this on L197.

P.9 line 201, Looking at the interpretation of Figure 3 and the MOMA gain thresholds, I would say the short periods with MOMA gains > 2.5 are mainly June and August. The majority of the dust days were in July and August with 2 days in September, we have added this information on L231.

P.9 line 215, "change the $PM_{2.5}/PM_{10}$ for each $PM_{2.5}$ source category" to "the reference $PM_{2.5}/PM_{10}$ for each $PM_{2.5}$ source category identified by MOMA". Done, L248.

P.10 line 224, change "source categories, and b)…" to "MOMA source categories, and b)…" We have changed "PM source categories" to "MOMA source categories throughout the manuscript.

P.10 line 224,   change "… b) normalized species concentrations measured for….." to "… b) normalized species concentrations measured at two Supersites for …"  We have clarified this on L268.

P.12 line 260,   change "Nr. " to "Number". Done.

P.14 line 303-305,   A.D. is not listed in the author list, remove? Also, C.C is missing from this list. Apologies, this was an error, we have corrected this and added C.C. to the contributions.

P.14 line 312-314,   Keep the abbreviation consistent, G.S.H should read G.H.  (see lines 303-305). D.E.W. is not an author in this manuscript, delete or include if missing. We have corrected this.

References

Weissert, L. F., Henshaw, G. S., Williams, D. E., Feenstra, B., Lam, R., Collier-Oxandale, A., Papapostolou, V., and Polidori, A.: Performance evaluation of MOMA (MOment MAtching) – a remote network calibration technique for PM2.5 and PM10 sensors, Atmospheric Meas. Tech., 16, 4709–4722, https://doi.org/10.5194/amt-16-4709-2023, 2023.